Optimization of multi-objective feature regression models for designing performance assessment methods in college and university educational reform

Qi Fengjun 1
Liu Zhenping 1 liuzhenping79@sina.com
Zhang Wenzheng 2
Sun Zhenjie 3
1 School of International Education, Nanning Normal University , Nanning, Guangxi , China
2 Krirk University International College , Bangkok , Thailand
3 Krirk University , Bangkok , Thailand
Asif Muhammad
Electronic publication date: 2025 Jun 5
Publication date: 2025
Volume: 11
Electronic Location ID: e2883
Received 2025 Jan 30; Accepted 2025 Apr 16
Copyright: © 2025 Qi et al.
Copyright year: 2025
Copyright holder: Qi et al.
License: This is an open access article distributed under the terms of the Creative Commons Attribution License, which permits unrestricted use, distribution, reproduction and adaptation in any medium and for any purpose provided that it is properly attributed. For attribution, the original author(s), title, publication source (PeerJ Computer Science) and either DOI or URL of the article must be cited.
License URL: https://creativecommons.org/licenses/by/4.0/

Keywords: Multi-objective regression, Performance appraisal, Goal stacking, Label-specific, Split box

Funding: National Social Science for Education Fund BGX230347 Funded by National Social Science for Education Fund-Special Projects for Western Areas: Research on the Current Status and Future Development of International Chinese Language Education Serving the Joint Construction of the “Belt and Road” between China and ASEAN (BGX230347). The funders had no role in study design, data collection and analysis, decision to publish, or preparation of the manuscript.

==============================
The evaluation of teacher performance in higher education is a critical component of educational reform, requiring robust and accurate assessment methodologies. Multi-objective regression offers a promising approach to optimizing the construction of performance evaluation index systems. However, conventional regression models often rely on a shared input space for all targets, neglecting the fact that distinct and complex feature sets may influence each target. This study introduces a novel Multi-Objective Feature Regression model under Label-Specific Features (MOFR-LSF), which integrates target-specific features and inter-target correlations to address this limitation. By extending the single-objective stacking framework, the proposed method learns label-specific features for each target and employs cluster analysis on binned samples to uncover underlying correlations among objectives. Experimental evaluations on three datasets—Education Reform (EDU-REFORM), Programme for International Student Assessment (PISA), and National Assessment of Educational Progress (NAEP)—demonstrate the superior performance of MOFR-LSF, achieving relative root mean square error (RRMSE) values of 0.634, 0.332, and 0.925, respectively, outperforming existing multi-objective regression algorithms. The proposed model not only enhances predictive accuracy but also strengthens the scientific validity and fairness of performance evaluations, offering meaningful contributions to educational reform in colleges and universities. Moreover, its adaptable framework suggests potential applicability across a range of other domains.

Introduction

With the progression of time, education has become increasingly significant in both social and family life, and public attention to the issue of educational quality has grown correspondingly (Carnoy, 2024). In constructing a teacher performance evaluation system in colleges and universities, the evaluation indices serve as the basis for assessing teachers’ overall performance. The rationality of the index system is directly linked to the intrinsic and extrinsic validity of the assessment (Russ-Eft, Preskill & Jordan, 2024). Teacher performance evaluation aims to examine how effectively teachers contribute to the achievement of institutional goals, and the design of performance evaluation indicators plays a pivotal role in this process. If these evaluation indices fail to provide accurate guidance for teachers’ work, it becomes challenging to align their performance with the broader objectives of the institution. Traditional performance evaluation methods often rely on a single indicator or a simple weighted average approach, which struggles to reflect teachers’ performance comprehensively and accurately (Li, Li & Luo, 2021). Such methods overlook the complexity and diversity inherent in teachers’ work, resulting in evaluation outcomes that lack scientific rigor and fairness.

Multi-target regression (MTR) (Yamaguchi & Yamashita, 2024) is an advanced statistical and machine-learning technique that uses a set of shared input variables to predict multiple dependent response variables. Existing methods within this framework can generally be classified into two main categories: problem transformation methods and algorithmic adaptation methods. Problem transformation methods typically convert a multi-objective regression problem into a series of single-objective regression problems or a combinatorial optimization problem. In contrast, algorithmic adaptation methods design or modify algorithms to accommodate the characteristics of multi-objective outputs.

In educational research, particularly in the evaluation of teacher performance in higher education, assessing students’ educational indicators across multiple dimensions represents a typical multi-objective learning scenario. Academic performance is influenced by factors such as students’ foundational knowledge, study habits, and teaching quality, while elements like family background, social environment, and school management shape behavioral performance. As a result, each objective is typically linked to its subset of characteristics. Consequently, when conducting performance assessments of educational reforms in higher education, it is essential to comprehensively consider a range of educational indicators and thoroughly analyze their relationships and interactions to devise more holistic and effective reform strategies. The multi-objective regression task presents significant challenges in practical applications. Since multiple targets need to be predicted simultaneously for each sample, accurately modeling the complex relationships between input features and each output target, as well as effectively exploring and leveraging the potential correlations between target variables, has become a pressing issue (Wei et al., 2024a). Most current multi-objective regression methods rely on a shared input space to predict all output targets. However, this approach may overlook the fact that each target could have its own distinct feature space. In practice, different output targets may be influenced by different input elements, and these influences may be nonlinear and nonmonotonic. As a result, using a common input space for prediction may fail to capture the unique information associated with each target fully, ultimately reducing prediction accuracy. Furthermore, most multi-objective regression methods typically employ a single approach to learn and predict all objectives, particularly in complex input-output relationships; however, in scenarios such as evaluating educational reforms, where the relationship between input features and target variables is highly nonlinear and the interactions between different targets can be intricate, relying on a single method may be insufficient to capture these complexities, leading to suboptimal prediction performance.

To address the issues above, this article proposes a multi-objective regression method that combines target-specific features and target relevance to optimize performance assessment methods for educational reform in higher education. The specific contributions of this article are as follows:

(1) To uncover inter-target correlations and improve prediction accuracy, this article extends the original input space by incorporating new features. It also learns label-specific features related to each target using boosting techniques within the framework of single-target stacking.

(2) Target-specific features are constructed through cluster analysis of binned samples, thereby modeling the complex relationship between the input and output spaces. Inter-target correlations are revealed by selectively stacking single-target predictive values.

(3) The multi-objective feature regression model, Multi-Objective Feature Regression model under Label-Specific Features (MOFR-LSF), is developed based on label-specific features. To model the intricate input-output relationships, this article proposes a multi-objective feature regression approach that integrates label-specific features.

“Related Work” reviews the current state of research on multi-objective regression modeling, highlighting the strengths and weaknesses in existing studies related to the performance evaluation of educational reforms in higher education. “Methodology” introduces the objective stacking, K-means clustering-based split-box, and MOFR-LSF models developed in this work. “Experiments and Analysis” presents the experimental results and discusses the performance of the MOFR-LSF model, as well as the impact of the enhanced predictive performance on the assessment of educational reforms in higher education. Finally, “Conclusion” summarizes the findings, discussing the MOFR-LSF model constructed in this study and outlining future research directions.

Related work

MTR is an extension of the traditional regression model that predicts multiple consecutive targets within the same input space. Existing MTR methods can be classified into two main categories: problem transformation methods (PTM) (Chung & Gazzola, 2024) and algorithm adaptation methods (AAM) (Muresanu et al., 2024). PTM transforms a multi-objective regression problem into individual single-objective regression problems, each of which is solved using a traditional regression approach. In contrast, AAM adapts specific single-objective regression methods to address multi-objective regression problems directly.

In recent years, most MTR models have been developed based on linear regression techniques to explore inter-objective correlations. However, these models cannot often handle high-dimensional inputs and the nonlinear relationships that may exist between multiple objectives (He, Sun & Xie, 2024). The work presented in del Barrio, González Sanz & Hallin (2024) introduces the Multiple-Output Regression with Task Structure (MROTS) model, which leverages the covariance structure of latent model parameters and the conditional covariance structure of the output space. It extends the multivariate regression model by incorporating covariance estimation and linear relationship multitask learning. However, MROTS lacks theoretical support for handling nonlinear regression tasks (Sevilla-Salcedo et al., 2024). Additionally, Zhang et al. (2024) proposes a linear regression model called calibrated multivariate regression (CMR) to address varying levels of noise across different tasks.

To address the nonlinear relationships between input elements and output targets, the Output Kernel Learning (OKL) algorithm (Zinage, Mondal & Sarkar, 2024) learns a semi-deterministic similarity matrix for multiple targets. It is capable of handling these nonlinear relationships. However, it has certain limitations. Specifically, the OKL algorithm is not fully comprehensive in mining inter-objective correlations, particularly when it comes to capturing negative correlations between objectives (Wei et al., 2024b). This limitation is especially critical in the performance evaluation of educational reforms in higher education, where the effects of reforms may involve multiple interrelated and potentially conflicting objectives, such as balancing improvements in academic performance with reducing student burden. In Saeed et al. (2024), the clustered multi-target learning (CMTL) model is proposed, which assumes that all tasks can be clustered into disjoint groups. This model explores the correlation between targets by clustering tasks and learning the underlying structure from the training data. However, CMTL also has its drawbacks. The algorithm requires the number of clusters to be specified in advance, a condition that poses challenges in the context of performance evaluation for educational reforms in higher education (Gill et al., 2025). Since such assessments often involve complex and dynamic factors, accurately determining the number of clusters beforehand is difficult, which can undermine the accuracy and reliability of the assessment results.

More recently, a Flexible Clustered Multi-Target Learning (FCMTL) regression algorithm has been introduced in Yang et al. (2024) to overcome some of the limitations of CMTL by using flexible clustering. The FCMTL algorithm identifies representative tasks to learn the cluster structure, yet it still faces challenges. The representative tasks in FCMTL are heuristically determined and lack general applicability, making them unsuitable for broader use. This implies that applying FCMTL to the performance evaluation of educational reforms in higher education may require extensive context-specific adaptation and optimization. This not only increases the complexity of implementation but also risks affecting the stability and consistency of the evaluation results.

Furthermore, since both multi-label classification and multi-objective regression share the core challenge of improving prediction accuracy by exploring dependencies between multiple objectives, Dowlatshahi & Hashemi (2024) adapts the learning approach of multi-label classification to multi-objective regression to investigate how the multi-label learning technique can be successfully applied in multi-objective regression problems. In Saidabad et al. (2024), Xia et al. (2022), Zhang & Chen (2024), this approach is further improved by mitigating the impact of chain order on the objectives. However, the discrepancy in the value of additional features between the training and prediction sets when using these methods can result in a significant degradation of prediction performance.

In summary, there are numerous challenges and limitations in designing performance assessment methods for university education reforms based on multi-objective feature regression model optimization. These challenges include the difficulty of comprehensively capturing inter-objective correlations, the requirement to specify the number of clusters in advance, and the heuristic nature of the representative task assumptions. Furthermore, current multi-objective regression methods predominantly focus on mining correlations between objectives and addressing the complex relationships between inputs and outputs. Most of these methods learn from a shared feature space to create a unified model that aims to predict all regression objectives simultaneously, which is insufficiently flexible. Therefore, these limitations must be carefully considered when designing performance assessment methods for educational reforms in higher education, necessitating the exploration of more suitable assessment methods and tools.

Methodology

We introduced a single-objective stacking framework in which label-specific features, closely related to each performance objective, are learned and extracted for each specific objective. The MOFR-LSF model is constructed by utilizing binning and clustering analysis strategies alongside the label-specific feature technique, ensuring that the overall assessment results remain balanced and consistent.

Specifically, we set D = (X, Y) as the training set of n samples, where X∈Rn∗m is the input space consisting of m feature vectors (X1,X2,…,Xj,…,Xm), and Y∈Rn∗d is the output space consisting of d target vectors (Y1,X2,…,Yi,…,Yd). (xi,yi) denote that a sample contains m-dimensional input variable (x1i,…,xmi) and d-dimensional output variable (y1i,…,ymi).

Target stacking

This section elaborates on the fundamental architecture of the MOFR-LSF model, which combines multi-output feature regression with a stacking framework. The model is designed to capture the unique input-output relationships for each education performance indicator and optimize predictions through target correlation analysis. By introducing target-specific features, the model is able to more accurately model the complex relationships among different indicators, laying a solid foundation for subsequent stacked regression.

We extend the single-target stacking (SSA) (Fisichella et al., 2015) method to multi-target learning by training d independent regression models using traditional regression techniques, as illustrated in Fig. 1. hj:X→R. hji:Dj\Dji→R.

Figure 1 The framework of SSA.

As shown in Fig. 1, for an unknown sample x, the prediction vector is first obtained through the model in the first stage:

(1) y^=(y^1,y^2,…,y^d).

The prediction vector is then expanded into the original feature vector, i.e.,:

(2) x′=(x1,…,xm,y^1,y^2,…,y^d).

Equation (2) shows a new input for the second stage model prediction and finally the final prediction is made with the second stage model hj′ to get the final prediction:

(3) y~=(y~1,...,y~d).

This approach considers the predicted values from the first stage of the remaining targets as the core idea for additional input variables, making it dependent on the estimates of the targets. To further alleviate inter-target dependencies, this article employs the cross-validation method to obtain out-of-sample estimates for the target variables across all training samples. The specific training process is illustrated in Fig. 2.

Figure 2 Training process.

K-means clustering for binning

Building on the target stacking approach, and to alleviate the challenge of constructing specific features for each target, this article adopts a split-box strategy based on K-means clustering of both targets and features for feature processing. First, the feature matrix X of the training set D undergoes column normalization to obtain X¯, the label part Y is unchanged and thus the normalized training set D¯=(X¯,Y) is obtained, K-means clustering is performed on each target with D¯i=(X¯,Yi) as input and thus binning is done to compute the sum of squares of the intra-cluster errors SSE at each K value:

(4) SSE=∑q=1k⁡∑p∈Cq⁡|p−mq|2

where Cq is the q-th cluster, p is the sample point in Cq, and mq is the mean of all samples in Cq. SSE (Sum of Squared Errors) represents the clustering error for all samples, indicating the quality of the clustering. From this, the SSE for different values of K (ranging from 2 to 20) is calculated for each target. The optimal number of clusters, i.e., the number of bins, for each target is then determined, allowing for the dataset to be partitioned accordingly after binning:

(5) Di={(Xi1,Yi1),...,(XiBi,YiBi)}

where i∈{1,...,m}. The split-box algorithm is carried out in the following steps.

(1) The feature part of dataset D, X=(X1,...,Xm), is normalized by column X¯=(X¯1,…,X¯m) to ensure that all feature values are within the same scale so that subsequent clustering algorithms can work more efficiently. The normalized dataset is recorded as D¯=(X¯,Y). The labeling part remains unchanged and serves as the basis for subsequent evaluation of the model performance.

(2) The dataset D¯=(X¯,Y) was used as input for the subsequent clustering algorithm. For each target variable i∈{1,...,m}, perform the following steps for binning.

For each target variable, K-means clustering of K=Bi is performed on the normalized dataset D¯i=(X¯,Yi) corresponding to the current target variable starting from a preset starting number of clusters s. The number of clusters is denoted as K. The sum of squared intra-cluster errors after clustering is computed using Eq. (4) to evaluate the effectiveness of the clustering process. This calculation continues until the loop reaches the maximum number of clusters, which is set to 20.

After completing the clustering cycle for the current target variable, the number of clusters that minimizes the SSE is selected as the optimal number of clusters based on the trend of the SSE values. Using the records from the normalization process, the normalized dataset after binning is then converted back to the original data scale to obtain the original dataset corresponding to the current target variable after binning.

(3) End the cyclic processing of all target variables and complete the binning of the entire dataset to obtain the binned normalized dataset D¯i={(X¯i1,Yi1),…,(X¯iBi,YiBi)}.

Once the binning process is complete, it becomes possible to apply concepts similar to multi-label taxonomies to identify specific features for each target.

Above, we discussed the K-means clustering strategy employed in the MOFR-LSF model. This strategy is used to identify and group samples with similar characteristics, simplifying model complexity and improving prediction efficiency. By selecting an appropriate number of clusters K, the model can more accurately capture the underlying structure of the dataset, providing strong support for feature selection and weight allocation in subsequent stacked regression.

Multi-objective feature regression model under label-specific features

Next, we will introduce the stacked regression component of the MOFR-LSF model. This component further enhances prediction performance by integrating the prediction results of multiple base learners and exploiting the correlations among targets. By introducing a target correlation threshold t, the model is able to screen out and stack prediction values that are highly correlated with the current target, thereby avoiding the interference of noise information. The MOFR-LSF framework, as shown in Fig. 3, is based on the concept of single-target stacking and the split-box technique. The data flow begins with the input features, which first undergo a preprocessing step to extract features specifically relevant to each educational performance indicator (i.e., label). These label-specific features are then used as inputs to the stacked regression framework, where each base learner is trained on a specific performance indicator. At the stacking layer, the model leverages the correlations among targets to generate the final predictions by weightedly combining the predictions of the base learners. In this process, the model not only considers the uniqueness of each label but also fully exploits the correlations between labels, thereby improving the accuracy of the predictions.

Figure 3 Framework of MOFR-LSF.

In this article, we extend the original feature space by computing initial predicted values before learning the label-specific features for each target. These initial predicted values are subsequently used as additional features.

Then, for each target Yj find features related to it XTR′, and construct label-specific features based on these target-related features. We use the residuals of each iteration to describe the local features as additional features to reflect the local information in the feature space:

(6) g0(XTR′)=fq(XTR′;Yj)

(7) rtj=−δl(Yj,gt−1(XTR′))δgt−1(XTR′)

(8) gt(XTR′)=gt−1(XTR′)+fq(XTR′;rtj).

Equation (6) is mainly used to set the initial condition, which is determined based on the basic learner of linear regression fq. In contrast, the function of Eq. (7) is to calculate the negative gradient value of the current model, which we treat as an estimate of the residuals. During this calculation, the squared error loss function is employed to assess the difference between the predicted and actual values:

(9) l(Yj,gt−1(XTR′))=12(Yj−gt−1(XTR′))2.

Experiments and analysis

In this section, we evaluate the performance of the proposed MOFR-LSF model by comparing it with four multi-objective regression algorithms: MROTS (del Barrio, González Sanz & Hallin, 2024), OKL (Zinage, Mondal & Sarkar, 2024), Sparse Structure Transfer (SST) (Fisichella et al., 2015), and Multilayer Multi-objective Regression (MMR) (Toktas, Ustun & Tekbas, 2019). Among these, MMR is a method capable of modeling both the internal correlation between objectives and the nonlinear input-output relationship within a single framework through robust low-rank learning. To demonstrate the generalizability and reliability of the performance assessment method for university education reform, optimized using the multi-objective feature regression model, we conduct experimental comparative analyses on three public datasets: EDU-REFORM (https://werd.stanford.edu/database), PISA (https://www.oecd.org/en/data/datasets/pisa-2018-database.html), and NAEP (https://catalog.data.gov/dataset/national-assessment-of-educational-progress-2016-1fcb4). The EDU-REFORM dataset reflects the effects and challenges of educational reforms in colleges and universities. The PISA dataset primarily examines the relationship between student performance and various educational factors, while the NAEP dataset provides insights into the relationship between student performance and educational policies and practices.

Experimental setting

The experiments in this article are conducted using the following experimental environment: a CPU of Intel® Core™ i7-11700F @ 2.50 GHz, dual NVIDIA GeForce GTX 1080 Ti GPUs, Python 3.8, PyTorch 1.8.1, Torchvision 0.9.1, CUDA 11.4, and CUDNN 6.0.

To determine the optimal number of clusters K in K-means, we adopted a method based on the sum of squared errors (SSE) within clusters. Specifically, we performed K-means clustering for each target variable, gradually increasing the number of clusters K from 2 to 20, and selected the number of clusters that minimized the SSE as the optimal one. Regarding the threshold t for target correlation, to avoid the model performance being degraded by predictions of targets with low correlation, we set it to a high value of 0.9, ensuring that only predictions highly correlated with the current target are stacked. Such adjustment strategies contribute to enhancing the model’s prediction performance and stability. Additionally, the experiments set the ratio parameter r to control the number of target-specific features. Specifically, a larger value of r leads to a higher value of K for each target’s sub-box, resulting in more centroids and higher-dimensional target-specific features. Based on the LIFT algorithm, the ratio r is set to 0.1 in this article.

A threshold t is used to measure the degree of consideration of target relevance. A higher value of t indicates fewer predicted values for the second-layer target predictions in the model stacking, thereby enhancing the relevance between targets. To prevent the predictions of weakly correlated targets from negatively affecting model performance, the threshold t is set to 0.9 in this study.

Evaluation criteria

Multi-objective regression is typically evaluated using the average relative root mean square error (aRRMSE), where the relative root mean square error (RRMSE) is defined as follows:

(10) RRMSE(h,Dtest)=∑(X,Y)∈Dtest⁡(Yi−Y^i)2∑(X,Y)∈Dtest⁡(Yi−Y¯i)2

where h is a multi-objective regression model, Dtest is the test set, (X,Y) is the test sample, Y^ is the predicted value of the ith objective of the test sample, and Y¯ is the mean value of the ith objective of the training set D. The smaller the value of RRMSE(h,Dtest) represents the better performance of the model h.

K-fold cross-validation is employed to compute the RRMSE for each fold of the dataset. The final RRMSE value is then obtained as the mean of the RRMSE values across all folds.:

(11) aRRMSE(h,D)=1k∑i=1k⁡RRMSE(hi,Di)

where D is the training set, and Di is the i-fold data after k-fold crossover of the training set D, which is used as the test set for the current cross-validation. hi is the model obtained by utilizing the remaining k-1 fold data as the training set. The smaller value of aRRMSE(h,D) represents the better performance of the model h.

Comparative analysis

MROTS, a multi-output regression model, leverages the covariance structure of latent model parameters and the conditional covariance structure of the output space, exhibiting certain similarities to MOFR-LSF in addressing multi-target regression problems and thus serving as a benchmark for comparison. Furthermore, methods such as OKL, SST, and MMR are also representative algorithms in the field of multi-target regression. Each of these methods has its unique characteristics: OKL excels in handling nonlinear relationships, SST has a research foundation in the single-target stacking framework, and MMR is capable of modeling both the internal correlations among targets and nonlinear input-output relationships through robust low-rank learning within a single framework. Therefore, this section will use these four models as our comparison models. Figure 4 illustrates the performance of the aRRMSE for the algorithms proposed in this article and for the comparison algorithms MROTS, OKL, SST, and MMR across three datasets. Figure 4D presents the average aRRMSE performance metrics for each algorithm across four datasets. From this, it can be observed that the average aRRMSE values are 0.87 for MROTS, 0.78 for OKL, 0.79 for SST, and 0.75 for MMR. All of these values are higher than the aRRMSE of the MOFR-LSF model presented in this article, indicating that the performance of the proposed model is superior.

Figure 4 aRRMSE comparison on different data sets.

(A) EDU-REFORM dataset; (B) PISA dataset; (C) NAEP dataset; (D) Average aRRMSE across EDU-REFORM, PISA, and NAEP datasets.

When dealing with datasets with complex nonlinear relationships, multilayer perceptrons (MLP) (Almeida, 2020) exhibit strong fitting capabilities. However, in terms of capturing correlations among targets, as shown in Fig. 5, they may not be as refined as the MOFR-LSF model. This is because the MOFR-LSF model, through its stacked regression framework and the construction of label-specific features, may more accurately predict different educational performance indicators. Additionally, the transformer-based model (Shojaee et al., 2023; Li et al., 2025; Huang et al., 2025) has advantages in processing sequential data and capturing long-range dependencies. However, in the context of evaluating the performance of higher education reform, the data does not fully exhibit sequential characteristics. Therefore, the MOFR-LSF model may be more suitable for capturing static correlations among targets.

Figure 5 Comparison with aRRMSE of multi-output regressors.

To highlight the features that contribute most significantly to prediction accuracy, we computed the SHAP values and permutation importance of each feature within the MOFR-LSF model. The results indicate that, across the three datasets—EDU-REFORM, PISA, and NAEP—students’ foundational knowledge and the quality of teaching have notable impacts on predicting academic performance. Conversely, other features such as family background and social environment are more critical in predicting behavioral outcomes. These findings not only validate the effectiveness of our feature selection method but also provide valuable reference information for evaluating the performance of higher education reform initiatives.

The above experiments involve statistical tests of multiple independent trials. However, as the number of comparisons increases, so does the probability of detecting significant differences by chance. The Bonferroni-Dunn test is a post-hoc method for multiple comparisons that mitigates the overall risk of Type I error by adjusting the significance level for each individual comparison. In the context of performance evaluation of educational reforms in higher education, different algorithms may exhibit varying performance across different datasets. The Bonferroni-Dunn test allows for a more precise identification of which algorithmic differences are statistically significant. To present the variability between the compared algorithms more intuitively, this article first calculates the critical difference and then illustrates it graphically. The critical difference is the minimum required difference in average rankings for two methods to be considered significantly different, where k is the number of comparison algorithms and N is the number of datasets.

Therefore, we set k = 9 and N = 18, and the critical value table of the Bonferroni-Dunn test at a = 0.05 is obtained as qa=1.506, at which time CD = 0.7954 can be obtained according to CD=qak(k+1)6N.

According to Fig. 6, the MOFR-LSF algorithm proposed in this article demonstrates a clear performance advantage. Specifically, a significant difference between two algorithms is considered when the difference in their average rankings exceeds 1.12395516. In Fig. 6, by comparing the critical difference (CD) values across the three datasets with the algorithm rankings, it is evident that the MOFR-LSF algorithm is positioned at the far right of the graph, with its average ranking difference being significantly larger than the CD value. This result highlights the superior performance of MOFR-LSF compared to algorithms such as MROTS, OKL, SST, and MMR, offering a more accurate and effective tool for evaluating the performance of educational reforms in universities.

Figure 6 The average ranking of each comparison algorithm.

(A) EDM database. (B) PISA database. (C) NAEP database.

Additionally, from Fig. 6, it is apparent that MOFR-LSF shows only a slight difference in performance compared to MMR but a significant difference when compared to OKL and SST. This suggests that considering target features alone is insufficient. Only by combining both target-specific features and target relevance can the performance of the algorithm be effectively enhanced. This article primarily focuses on enhancing the accuracy and scientific rigor of performance evaluations for higher education reform initiatives. As the size of the dataset increases, the training and prediction time of the model may also grow, necessitating trade-offs and optimizations in practical applications.

Ablation experiments

To verify whether label-specific features contribute to improving the prediction performance of the multi-objective feature regression model, ablation experiments are conducted with label-specific features serving as control variables. The experiment that follows the same process as MOFR-LSF but excludes label-specific features is designated as E1. Figure 7 presents the prediction results of MOFR-LSF and E1 using the evaluation metric aRRMSE. As shown in Fig. 6, the prediction performance of MOFR-LSF surpasses that of E1 on both the PISA and NAEP datasets. Label-specific features are capable of capturing the unique input-output relationships for each target, aiding the model in more refined modeling and prediction. Additionally, these features can reveal the correlations among targets, further enhancing the model’s expressive power and generalization ability. Therefore, label-specific features are one of the key factors contributing to the performance improvement of the MOFR-LSF model.

Figure 7 Label validity.

In the MOFR-LSF framework, we introduce the sparse integration method to address the complex associations between input features and output targets. To verify the effectiveness of this sparse integration, we use it as a control variable in ablation experiments. Specifically, this article compares MOFR-LSF with support vector regression (SVR-LSF) (Huang & Lian, 2023) and random forest (RF-LSF) (Mzobe et al., 2020; Jiang et al., 2024; Li & Xing, 2025), both of which do not utilize sparse integration. Figure 7 presents the aRRMSE values of SVR-LSF, RF-LSF, and MOFR-LSF across different datasets. As shown in Fig. 8, the prediction results of MOFR-LSF outperform those of the other two methods on all datasets, demonstrating superior performance. This result further indicates that the introduction of the sparse integration method in this study effectively addresses the complex associations in multi-objective relationships, thereby enhancing the performance evaluation of educational reforms in universities.

Figure 8 Comparison of prediction performance of the algorithm for each target under different data sets.

(A) EDU-REFORM dataset; (B) PISA dataset; (C) NAEP dataset.

Discussion

Traditional performance assessment methods for college education reform often rely on a single indicator or a simple weighted average, a method that is difficult to fully reflect teachers’ multidimensional performance. The MOFR-LSF model, on the other hand, by mining the correlation between targets and constructing target-specific features, is able to capture all aspects of teacher performance in a more comprehensive way, thus providing more accurate assessment results. In addition, the model adopts a split-box method based on K-Means clustering to construct target-specific features, which effectively reduces redundant features and improves the expressive power of the model. The experimental results demonstrate that the MOFR-LSF model is capable of capturing the unique input-output relationships for each educational performance indicator. This implies that it can provide customized predictions and evaluations tailored to different education reform objectives. Such customized predictions furnish decision-makers with more precise information, aiding them in understanding the specific impacts of various reform measures on different performance indicators. Furthermore, the model integrates the predictions of multiple base learners through the stacked regression framework and leverages the correlations among targets to optimize overall prediction performance. This means that decision-makers can obtain a more comprehensive and accurate performance prediction, enabling them to weigh the pros and cons of different reform measures more scientifically and make more informed decisions.

In summary, the MOFR-LSF model has significant application advantages in the performance assessment of educational reform in colleges and universities, which not only improves the comprehensiveness and accuracy of the assessment but also enhances the scientificity and fairness of the assessment. The application of the model helps promote the development of education reform in colleges and universities, promotes the personal growth of teachers, and improves education quality.

Conclusion

This article proposes an innovative multi-objective regression method that extends the single-objective stacking framework to learn label-specific features relevant to each objective. Simultaneously, the binned samples are analyzed through clustering to construct target-specific features, thereby revealing the correlations between the targets. This approach not only improves prediction accuracy but also enhances the comprehensiveness and precision of the model in evaluating the performance of university education reform. Experimental analysis is conducted on three datasets, demonstrating significant improvements in prediction accuracy as the model complexity increases. However, this increased complexity may reduce the model’s interpretability.

Furthermore, when dealing with large-scale datasets, model training and prediction can become time-consuming and resource-intensive. Therefore, enhancing computational efficiency while maintaining model performance remains a key challenge. In future work, we aim to explore techniques such as distributed and parallel computing to accelerate the model’s training process and incorporate rule-based or tree-based methods to construct more interpretable multi-objective regression models.

Supplemental Information

Supplemental Information 1 Code.

Additional Information and Declarations

Competing Interests

The authors declare that they have no competing interests.

Author Contributions

Fengjun Qi conceived and designed the experiments, performed the experiments, authored or reviewed drafts of the article, and approved the final draft.

Zhenping Liu performed the experiments, performed the computation work, prepared figures and/or tables, authored or reviewed drafts of the article, and approved the final draft.

Wenzheng Zhang conceived and designed the experiments, analyzed the data, performed the computation work, prepared figures and/or tables, and approved the final draft.

Zhenjie Sun analyzed the data, prepared figures and/or tables, authored or reviewed drafts of the article, and approved the final draft.

Data Availability

The following information was supplied regarding data availability:

The World Education Reform Database is available at Harvard Dataverse: Bromley, Patricia; Kijima, Rie; Overbey, Lisa; Furuta, Jared; Choi, Minju; Santos, Heitor; Song, Jieun; Nachtigal, Tom; Yang, Marcia, 2024, “World Education Reform Database (WERD)”, https://doi.org/10.7910/DVN/C0TWXM, Harvard Dataverse, V3.

The PISA 2018 Database is available at: https://www.oecd.org/en/data/datasets/pisa-2018-database.html.

The National Assessment of Educational Progress, 2016 (NAEP) data is available at: https://catalog.data.gov/dataset/national-assessment-of-educational-progress-2016-1fcb4.

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
