# Peer review of "Optimization of multi-objective feature regression models for designing performance assessment methods in college and university educational reform"

_PeerJ Computer Science, doi:10.7717/peerj-cs.2883_

## Round 0.1 · original submission · Major Revisions

Dear authors, the experts has now commented on your manuscript and you will see that they are recommending a number of changes to be incorporated before we reconsider it. I do agree with them and advise the following
1. Justify the need of the research.
2. Validation of the study is needed .
3. The technical language needs improvement.
4. Explain the practicality of your research,eg where it could be useful?

Reviewer 1 ·

Basic reporting

• Some sentences are overly complex and difficult to read.

Experimental design

• The manuscript states that the MOFR-LSF model extends the feature space by computing initial predicted values before learning label-specific features. However, the criteria for feature selection remain unclear. Include a feature importance ranking using techniques like SHAP values or permutation importance in the results section to highlight which features contribute the most to prediction accuracy.
• The paper compares MOFR-LSF with MROTS, OKL, SST, and MMR, but lacks an explanation of why these models were chosen as baselines. Provide a table summarizing each baseline model's key attributes and how they compare to MOFR-LSF in terms of complexity, feature engineering, and computational efficiency.
• The manuscript does not mention how hyperparameters were tuned, particularly for the number of clusters (K) in K-Means and the threshold t for target relevance.
• The study presents an ablation analysis where label-specific features are removed (E1 model), but it lacks an analysis of why these features contribute positively to model performance.

Validity of the findings

• The introduction of K-Means clustering and stacking regressions increases computational load. Include a runtime comparison table for MOFR-LSF versus baselines and discuss its scalability on larger datasets.
• Implement Bayesian regression or Monte Carlo Dropout to estimate uncertainty and present confidence intervals for predictions.
• Simplify key methodological explanations, particularly in Sections 3.1–3.3, and ensure clarity in model equations.

Additional comments

This manuscript presents a novel Multi-Objective Feature Regression Model under Label-Specific Features (MOFR-LSF) for optimizing performance assessment methods in educational reform. While the approach is interesting and demonstrates promising results, some concerns need to be addressed regarding model justification, feature selection process, and result interpretation.

Reviewer 2 ·

Basic reporting

The proposed MOFR-LSF model presents a well-structured framework for improving educational performance assessments. However, some technical aspects of the modeling process and result validation need further refinement.
The manuscript briefly describes how label-specific features are used, but lacks a step-by-step breakdown. Provide a flowchart explaining data flow from input to final prediction.
The manuscript mentions that feature selection is done through clustering, but does not discuss how collinearity among input features is addressed.
Real-world examples of how MOFR-LSF improves decision-making are missing.

Experimental design

The study does not discuss training time, inference time, or GPU utilization.
Include a table comparing computational costs across all models.

Validity of the findings

Compare Against Deep Learning Methods (Section 4.3). Add a comparison with neural network-based multi-output regressors like Multi-Layer Perceptron (MLP) or Transformer-based models.
Apply VIF (Variance Inflation Factor) analysis or Principal Component Analysis (PCA) to reduce feature redundancy.

---

## Round 0.2 · accepted · Accept

Dear authors

Thank you for resubmitting the paper after updating it. The experts have commented on the revised version of your paper, and I am pleased to inform you that they are now in favor of accepting your article. Congratulations! And thank you for your fine contribution.

Reviewer 1 ·

Basic reporting

no comment

Experimental design

no comment

Validity of the findings

no comment

Reviewer 2 ·

Basic reporting

All the issues I pointed out after the first review have been solved

Experimental design

All the issues I pointed out after the first review have been solved

Validity of the findings

All the issues I pointed out after the first review have been solved